# Comparative Toxicological Analyses of Traditional Matrices and Blow Fly Larvae in Four Cases of Highly Decomposed Human Cadavers

**DOI:** 10.3390/insects15070500

**Published:** 2024-07-04

**Authors:** Michela Peruch, Maria Buffon, Zlatko Jakovski, Chara Spiliopoulou, Riccardo Addobbati, Martina Franzin, Paola A. Magni, Stefano D’Errico

**Affiliations:** 1Department of Medical Surgical and Health Sciences, University of Trieste, 34127 Trieste, Italy; michela.peruch@studenti.units.it (M.P.); maria.buffon@studenti.units.it (M.B.); 2Medical Faculty, Institute for Forensic Medicine, Criminology and Medical deontology, University of St. Cyril and Methodius, 1000 Skopje, North Macedonia; drjakovski@gmail.com; 3Department of Forensic Medicine and Toxicology, School of Medicine, National and Kapodistrian University of Athens, 75 Mikras Asias, 115 27 Athens, Greece; chspiliop@gmail.com; 4Institute for Maternal and Child Health, IRCCS “Burlo Garofolo”, 34137 Trieste, Italy; riccardo.addobbati@burlo.trieste.it (R.A.); martina.franzin@burlo.trieste.it (M.F.); 5School of Medical, Molecular & Forensic Sciences, Murdoch University, Murdoch, WA 6150, Australia

**Keywords:** entomotoxicology, putrefaction, calliphorid, crime scene, postmortem

## Abstract

**Simple Summary:**

Determining the time and cause of death in forensic investigations is challenging when remains are highly decomposed, making traditional toxicological samples unreliable. Entomotoxicology, a controversial and underinvestigated field in forensic science, uses insect specimens from cadavers as alternative toxicological samples. Research has shown that insects feeding on decomposing tissues can serve as substrate for toxicological analyses, which can detect drugs, toxins, and elements even at low concentrations. Nevertheless, the correlation of drug concentrations between insects and conventional matrices is controversial due to unknown factors affecting drug metabolism. This paper examines four real cases where human cadavers were in advanced decomposition, and toxicological analyses were performed on both insect samples and available matrices. The findings provide insights into the correlation between larvae and human specimen results, adding to the limited literature on entomotoxicology in real cases. Additionally, this paper offers guidelines for collecting and preserving entomological evidence at crime scenes and during autopsies for use in entomotoxicological analyses. This advancement shows promise in enhancing forensic investigations, particularly in cases where traditional methods are inapplicable or require additional validation.

**Abstract:**

In forensic investigation, determining the time and cause of death becomes challenging, especially in cases where the remains are found in advanced decomposition, rendering traditional toxicological samples unavailable or unreliable. Entomotoxicology, an emerging methodology within forensic science, leverages insect specimens collected from cadavers as alternative toxicological samples. Several laboratory and field research studies have highlighted the efficacy in detecting various drugs, toxins, and elements absorbed by insects feeding on cadaveric tissues, even at low concentrations. However, correlation studies between drug concentrations in conventional matrices and insects remain controversial due to unknown factors influencing drug metabolism and larval feeding activity. This paper presents four real cases in which human cadavers were discovered in advanced stages of decomposition, and toxicological analyses were performed on both insect samples and available matrices. The results presented complement the scant literature currently available on the application of entomotoxicology in real cases, providing insights into the correlation between larvae and human specimen results. Furthermore, guidelines to collect and preserve entomological evidence at the crime scene and during the autopsy for use in entomotoxicological analyses are provided. This advancement holds promise in aiding forensic investigations, particularly in cases where traditional methods cannot be applied or require supporting data for further validation.

## 1. Introduction

In forensic pathology investigations, the estimation of the time and the understanding of the cause of death are critical factors for determining the circumstances surrounding a person’s demise and guiding legal and investigative processes. Accurate determination of these factors not only aids in reconstructing the sequence of events but also in verifying or refuting witness statements, evaluating alibis, and guiding further investigative steps. This can be particularly challenging in cases where bodies are in an advanced stage of decomposition. In such instances, the typical signs of death used to estimate the time of death (postmortem interval, PMI) cannot be used, and traditional toxicological samples like tissues or body fluids may not be available or their analysis results could be unreliable. However, when remains decompose in a terrestrial temperate environment without any or with limited physical restrictions, they are rapidly colonized by necrophagous insects that feed on the cadaveric tissues. The insect species and their development compared to the micro/macro environmental temperature can be used to estimate the cadaver’s PMI (forensic entomology) [1], and the application of toxicology methods to the insects can be used to ascertain the presence of exogenous substances in the body, including drugs, elements, and toxins. This methodology, known as “entomotoxicology”, has emerged as promising within forensic science, enhancing the precision and reliability of toxicological determinations, especially in cases involving advanced decomposition where traditional approaches may be compromised [2].

Entomotoxicology has a relatively short history, with early research dating back to the 1970s [3] and 1980s [4]. Since then, numerous studies have demonstrated the presence of drugs and toxins in insect samples collected from decomposing bodies [5,6]. The efficacy of drug detection from insect samples hinges on the employed extraction techniques and analytical methods. Chromatography coupled with mass spectrometry (GC-MS) is regarded as the gold standard owing to its sensitivity and accuracy; however, several other methods have been applied, such as high-performance liquid chromatography (HPLC) and radioimmunoassay (RIA) [7]. Nonetheless, challenges persist in correlating drug concentrations between substrates and insects. These challenges include the unpredictable or understudied bioaccumulation of the toxicant in the insect body and its biomagnification throughout the food chain. Factors such as the toxicant type and quantity, insect species and age [8], and environmental conditions affecting metabolism [9] determine whether the toxicant can be assimilated, digested, absorbed, sequestered, metabolized, or excreted. Additionally, the pharmacokinetics of toxicological substances in cadaver bodies depend on variables such as substance type and quantity, exposure duration, and body size and condition. Moreover, the thanatomicrobiome, which includes bacteria responsible for putrefaction, can degrade toxicants, leading to the loss of analytes and metabolites [10], further complicating drug concentration correlation. Despite this, studies have shown promising results in detecting drugs from insects, even at low concentrations. Pien et al. demonstrated that detection and quantification of the drug Nordiazepam and its metabolite Oxazepam could be carried out from a single larva and puparia down to the picogram level [11]. 

Matrices of entomotoxicological interest generally include immature carrion insects (larvae, full puparia) and their remains (pupal cases, beetle exuviae). Additionally, toxicological analyses can be performed on carrion beetle faecal material (dermestid frass) [12] and direct predators of carrion insects. Overall, third instar blowfly larvae and puparia are the most common samples used in entomotoxicological analyses [5], because their presence is generally associated with advanced stages of decay, and the pupal stage represents the longest period in the blowfly immature life [13]. Furthermore, following the eclosion of the adult fly, pupal cases can persist in the environment for hundreds of years, allowing for toxicological analyses of samples even of archaeological interest [14].

To date, scientific literature reports several studies focusing on the quantification of xenobiotics in conventional matrices (such as blood, urine, and vitreous humour) and parenchymatous organs. Nevertheless, the correlation between substance quantification in insects and cadaver organs is highly controversial [11,15,16,17]. For example, in an extensive study on 29 human cadavers suspected to have died due to poisoning, the authors did not observe correlation between drug concentrations in human tissue and larvae samples [17]. On the contrary, other studies have shown a correlation between drugs in substrates and larvae samples [18,19]. In most of these studies, larvae were fed on minced meat without allowing the drug to undergo natural metabolic processes. Therefore, the findings and results may differ from those of actual poisoning cases and only a limited number of studies of forensic pathology importance are available.

Given the limited availability of entomotoxicology studies of forensic pathology significance [20], this research aims to address this gap. In this paper, four cases of human subjects found in an advanced state of putrefaction are reported, with toxicological analyses conducted on both insect samples and conventional matrices. Where possible, comparative analyses between insects and conventional samples were conducted.

The findings of this study provide insights into the utility, accuracy, and reliability of entomotoxicological analyses in cases of advanced decomposition, thereby enhancing the overall investigation process.

## 2. Materials and Methods

### 2.1. Case Histories

The following case report includes 4 unrelated subjects discovered several weeks after death and analysed at the Institute of Forensic Medicine of the University of Trieste, Italy. The cases are heterogeneous both for age and sex of the subjects, as well as time of the year, environmental conditions, and presence of drugs in the surrounding environment. Details of each case are summarized in Table 1. It should be noted that in all of these cases, the Public Prosecutor in charge of the investigation requested an investigation regarding the causes and manner of death, but not an estimation of the time of death based on the entomological evidence. As a consequence, micro/macro-environmental data were not collected, and not all specimens underwent species identification.

#### 2.1.1. Case 1

At the beginning of October, the remains of a man in an advanced state of decay were discovered in his home. In the kitchen, the gas knobs were found to be open. Additionally, numerous blister packs of drugs were discovered, including zolpidem, gabapentin, duloxetine, levosulpiride, propantheline bromide/bromazepam, paracetamol/codeine phosphate, and many others.

The body exhibited advanced putrefactive phenomena in the colliquative phase, with the upper regions almost completely skeletonized. A plastic bag covered and contained the skull, jaw, and the cervical vertebrae, as well as a butane can and a large number of third instar blow fly larvae (*Chrysomya albiceps* (Wiedemann) (Diptera: Calliphoridae)). Blow fly larvae, full and empty puparia, and adults of the same species were also observed on the body surface and on the floor surrounding the body (Figure 1). A postmortem CT-scan performed to identify any sign of traumatic injury obtained a negative result. Toxicological analyses were performed on *Ch. albiceps* larvae collected from the body and surroundings, hair, and muscle samples. Based on circumstantial data and medico-legal examination, the estimated PMI was approximately 20–30 days.

#### 2.1.2. Case 2

At the end of August, the remains of a woman in an advanced state of decay were discovered in her apartment, approximately three weeks after she was last seen alive (eyewitness testimony). It was known that she was undergoing substitutive treatment with methadone at the local SerD (service for drug addiction). Many blister packs of drugs were found in her apartment, including quetiapine and rivotril, which were regularly prescribed, along with some empty bottles of methadone hydrochloride. Additionally, one blister pack of quetiapine and three bottles of methadone were found next to the corpse.

The body exhibited postmortem decomposition, with advanced putrefactive phenomena in the cromatic-colliquative phase. Initial stages of skeletonization and mummification were observed in certain parts of the body. No indications of external violence to the body could be identified. There was a large colonization by *Ch. albiceps*, with active feeding third instar larvae observed on the body surface, alongside post-feeding larvae in the surrounding environment (Figure 2). A toxicological investigation was conducted on larvae collected from the highest colonized area of the body and on a spleen sample taken during the autopsy.

#### 2.1.3. Case 3

At the end of September, a homeless man was found hanged by a rope wrapped around his neck, outside the parapet next to the guardrail of a street. His head was covered by a short-sleeved shirt, and his wrists were tied together by a long-sleeved shirt. Scotch tape was wrapped around his ankles. The deceased was discovered approximately one week after he was last seen alive. During that week, the weather conditions alternated between rainy and sunny days [21]. Following his identification, a medical report was found indicating a diagnosis of anxious–depressive syndrome with previous substance abuse. No information regarding ongoing medical therapies was available.

An external examination found no signs of external violence; however, a skin compression groove was noted along the right, anterior and left surface of the neck, becoming more superficial at the posterior surface. The body exhibited postmortem decomposition, with advanced putrefactive phenomena in the chromatic-colliquative phase. The scalp, face, neck and the superior region of the chest showed transformative phenomena such as corification. Third instar calliphorid larvae were observed on the entire body surface, especially on the upper body and right orbit (Figure 3). Toxicological investigations were performed on peripheral blood and larvae samples collected from the most colonized area of the body.

#### 2.1.4. Case 4

At the beginning of January, human remains were found in a wooded area. The body was partially disarticulated, with some limbs missing, and the skull was discovered a few meters away from the body (Figure 4). The remains exhibited partial skeletonization and transformative phenomena of mummification and saponification. The body was colonized by third instar calliphorid larvae, which were used for subsequent toxicological investigations. Medico-legal, anthropological, and odontological examinations confirmed the body belonged to a woman who had been missing for almost three months. Her identity was confirmed through the comparison of an orthopantomography report. The woman had been undergoing treatment with flurazepam, quetiapine, and lorazepam, although the prescription was dated more than a year prior, so it is unclear if the treatment was still in effect at the time of death.

### 2.2. Biological Samples for Toxicological Analyses

Specimens used for toxicological analyses were collected by forensic pathologists during scene investigation (entomological samples) or during the autopsy (human specimens).

In accordance with current forensic entomology guidelines [22], blow fly larvae were gently washed in MilliQ water to remove debris from the environment and human remains, and stored in a freezer (−20 °C) until analysis. Similarly, if available, human specimens were routinely collected and stored in a freezer (−20 °C) until analysis. Toxicological analyses were performed at the Advanced Translational Diagnostic Laboratory at the University of Trieste, School of Forensic Medicine. Ethics permission was not necessary as the toxicological analyses were requested by the prosecutor for judicial purposes.

### 2.3. Chemicals and Reagents

All chemicals and reagents were of analytical grade. MassTox Drugs of Abuse testing Mobile phases A, B, and rinsing solution, MassTox Drugs of Abuse Analytical column, 6Plus1 Multilevel Urine Calibrator SET, MassCheck Drugs of Abuse testing urine, MassTox Drugs of Abuse testing Internal Standard (consisting of deuterated compounds), MassTox Drugs of Abuse testing Enzyme solution set, MassTox Drugs of Abuse testing Precipitation reagent and Dilution buffer were purchased from Chromsystems Instruments & Chemicals GmbH (Munich, Germany) and used for the quantification of femoral venous blood, urine, larvae, liver, kidney, skeletal muscle, and spleen postmortem specimens. M3 Reagent, M3 solvents, calibrators Cal M3 and Tricocheck controls were purchased from Comedical srl (Trento, Italy) and used for the quantification of the hair sample. Dulbecco’s phosphate buffered saline was purchased from Euroclone (Milan, Italy). Formic acid and ammonium formiate were purchased from Sigma Aldrich (Milan, Italy).

### 2.4. Sample Preparation

For entomological samples, 3 mL of methanol were added to 1 g of larvae in a 7 mL tube, and specimens were blended to form a smooth homogenate using Bead Ruptor Elite (Omni International) with an optimized protocol (6 m/s, 3 cycles, 15 s of action, 5 s of pause). After centrifugation, 1 mL of the supernatant was dried under a gentle stream of nitrogen and then resuspended in 50 µL of phosphate buffer.

For human samples, 3 mL of methanol were added to 1 g of tissue derived from the liver, kidney, skeletal muscle, spleen, and hair and placed in a 7 mL tube. The specimens were blended to form a smooth homogenate using Bead Ruptor Elite (Omni International) with a specific protocol repeated 3 times (liver: 5 m/s, 2 cycles, 20 s of action, no pause; kidney: 4 m/s, 3 cycles, 10 s of action, 10 s of pause; skeletal muscle: 6 m/s, 1 cycles, 30 s of action, no pause; spleen: 4.5 m/s, 2 cycles, 10 s of action, 10 s of pause). After centrifugation, 1 mL of the supernatant was dried under a gentle stream of nitrogen and then resuspended in 50 µL of phosphate buffer.

The sample preparation differs depending on the matrix. For specimens where illicit drugs accumulate after metabolism, such as blood, skeletal muscle, and spleen, enzymatic hydrolysis by β-glucuronidase was not required. In contrast, other matrices such as urine, liver, kidney, and larvae, required enzymatic hydrolysis. Therefore, for the preparation of urine, liver, kidney, and larvae, 10 µL of internal standard mix (IS) and 40 µL of β-glucuronidase enzyme, whose efficiency of enzymatic hydrolysis was previously tested by Chromsystems Instruments & Chemicals GmbH (Munich, Germany), were added to 50 µL of sample. After brief mixing, samples were incubated for 2 h at 45 °C to allow enzymatic deconjugation. At the end of the incubation, 100 µL of precipitant reagent was added and, after vortexing, the sample was centrifuged for 5 min at 14,500 rpm. To 100 µL of supernatant, 150 µL of dilution buffer was added. The same preparation method was used for femoral venous blood, skeletal muscle, and spleen, without the enzymatic reaction step. The preparation of the hair sample was performed as per the instructions of the manufacturer (Comedical srl).

Screening and confirmatory analyses were performed with an HPLC Exion LC 2.0 (Sciex, Milan, Italy) coupled with a QTRAP 6500+ system (Sciex, Milan, Italy), equipped with an electrospray ion source (ESI) operating in positive and negative modes.

### 2.5. Forensic Toxicology Screening 

Simultaneous screening of over 700 illicit drugs was conducted using a workflow based on multiple reaction monitoring (MRM), followed by the acquisition of full-scan MS/MS data (Enhanced Product Ion (EPI)) and automated MS/MS library searching. Chromatographic separation was achieved by eluting mobile phase A (10 mM ammonium formiate) and B (0.05% formic acid in methanol) on a stationary phase Kinetex 2.6 μm Phenyl-Hexyl 100 A, LC Column 50 × 4.6 mm (Phenomenex, Milan, Italy). The injection volume was 15 µL. The analysis was carried out at a flow rate of 0.7 mL/min, with a gradient from 10% to 98% organic solvent over 8.5 min to separate all molecules. Acquisition in MRM was conducted in both positive and negative modes. The intensity threshold for performing automated MS/MS library searching was set at 40,000 cps, and target ions were excluded for 5 s.

### 2.6. Forensic Toxicology Confirmation 

Confirmatory analysis was performed on samples that had previously undergone screening tests. For confirmation of the identification of illicit drugs in urine, femoral venous blood, larvae, liver, kidney, skeletal muscle, and spleen samples, an LC-MS/MS method was employed according to the manufacturer’s instructions (Chromsystems Instruments & Chemicals GmbH, Munich, Germany).

For confirmation of the identification of illicit drugs in hair, LC-MS/MS analysis was conducted according to the manufacturer’s instructions (Comedical srl).

### 2.7. Data Processing and Statistical Analysis

Data processing and analysis were conducted using Analyst (version 1.7), Multiquant (version 3.0.2) and Sciex OS (version 3.0) software. Additionally, concentrations of illicit drugs derived from the confirmatory analysis were calculated by normalizing the response ratio of analytes with that of the IS and interpolating with the calibration curve obtained from blood. Since analyses were performed on aqueous extracts of specimens, an attempt was made to establish a “blood surrogate”.

## 3. Results

Table 2, Table 3 and Table 4 summarize the findings of the toxicological analyses for cases 1, 2, and 4, including quantitative analysis. Case 3 resulted as negative for all tested illicit drugs.

## 4. Discussion

Toxicological analyses are pivotal in determining the cause and circumstances of death in forensic cases, particularly those characterized by long postmortem intervals (PMIs), where body degradation may obscure critical elements necessary for thorough examination. This degradation process, exacerbated by environmental exposure, can alter typical matrices routinely used in toxicological analyses, such as blood, urine, and liver. Additionally, the feeding activity of necrophagous insects can further limit the availability of these matrices. Despite these challenges, insects can serve as a substitute or adjunct for conducting toxicological analyses, providing non-conventional matrices for examination.

To date, only a limited number of papers have been published describing case work involving toxicological and entomotoxicological examinations, in comparison to research studies in the field of entomotoxicology. Furthermore, some of these studies may be outdated as they are affected by limitations in analytical procedures or the use of obsolete techniques [23]. Such methods may not provide the improved sensitivity and precision offered by modern machinery, which boasts enhanced limits of detection and quantification. 

This paper presents findings from four cases with estimated PMIs ranging from one week to three months, exhibiting various stages of putrefactive changes, from discoloration to partial skeletonization, all showing colonization by necrophagous insects. In these cases, circumstantial evidence or knowledge of the deceased’s medical history necessitated toxicological profiling alongside routine autopsy procedures to achieve a comprehensive understanding of the causes of death. To enhance the investigative process of these cases, both insect and human samples were analyzed whenever possible. This approach enables a more comprehensive toxicological screening, addressing any potential issues related to postmortem redistribution (PMR) [24]. PMR is a phenomenon occurring after death, whereby certain substances can ‘move’ from one compartment (e.g., blood) to another (e.g., a peripheral organ), based on their chemical and physical properties. Consequently, data obtained from forensic toxicology analyses may vary depending on the time elapsed between death and sample collection. Thus, values for a particular substance may increase in one matrix while decreasing in another.

Case 1: This case exemplifies a complex suicide scenario. Based on circumstantial evidence, scene investigation, and postmortem CT-scan results, the cause of death was attributed to central and peripheral cardio-respiratory failure resulting from the toxic depressive-inhibitory effects of inhaling butane combined with mechanical asphyxia from confined space. However, none of the available matrices (pubic hair, muscle, or larvae) proved suitable for detecting volatile substances. Despite this limitation, toxicological analysis for common substances revealed the presence of cocaine, codeine, bromazepam, diazepam, and zolpidem, along with some metabolites. As highlighted by Stine Lund Hansen and collaborators [25], muscle tissue serves as a viable alternative matrix when blood samples are unavailable. Their study demonstrates a correlation between blood concentration and muscle concentration, with higher levels observed in muscle tissue. Consequently, none of the detected medications was deemed the primary cause of death.

Comparison of results obtained from muscle and hair samples indicates chronic intake of codeine and zolpidem, while the drug concentration detected in the muscle suggests acute intake before death. Considering the literature data [26] indicating that lethal codeine levels range from 1 mg/L to 8.8 mg/L in blood, the levels detected in the muscle are neither lethal nor do they imply the drug was consumed a significant amount of time prior to death. Conversely, the detection of bromazepam, diazepam, and their metabolites solely in muscle (not in hair) suggests acute consumption of these substances.

The most noteworthy discovery is that every substance detected in muscle and hair samples was also found in larvae samples, except for cocaine. This case represents the first in which this mixture of drugs has been detected in blow fly larvae, opening research avenues to assess if larvae may be more suitable for assessing acute intake rather than chronic use. The presence of cocaine traces solely in hair suggests prior substance use by the subject, without recent consumption before death.

Case 2: In this case, the subject was undergoing methadone treatment. The literature data show that the optimal matrix for toxicological investigation of methadone is the brain, yielding results consistent with those of blood samples [27]. Furthermore, it has been demonstrated that results obtained from the liver, spleen, and kidney are similar, although the concentrations found were higher than those obtained from the blood matrix [28]. Additionally, a recent study showed an average methadone PMR rate of 6%, with values ranging from −46% to +28%, while for its metabolite, EDDP, the average value was +25% (−36; +87%) [24]. According to the literature, liver methadone concentration in fatal intoxication cases ranges from 1.8 to 7.5 mg/kg, with an average concentration of 3.8 mg/kg [26]. In Case 2, the spleen methadone concentration was 27,699.0 ng/g, about seven times the average toxic threshold. Even considering the biases introduced by PMR and the fact that the matrix was different from the reference matrix, it is possible to assume that methadone intoxication was likely the leading cause of death in the subject under investigation. Additionally, the detection of methadone metabolite EDDP in the amount of 1990 ng/g in the spleen sample indicates a substance/metabolite ratio skewed towards the substance, consistent with acute ingestion prior to death. Moreover, the further central nervous system depressing effects by the benzodiazepine (7-aminoclonazepam) may have contributed to the fatality. 

Spleen analysis also detected the presence of quetiapine in the amount of 3693 ng/g. Although irrelevant to the cause of death, this result confirms that all substances and empty blisters found during scene investigation were consumed while the subject was alive.

Toxicological analysis performed on the blow fly larvae validated the qualitative results obtained with the spleen sample, albeit at lower concentrations. It is worth noting that, in this matrix as well, the methadone/EDDP ratio is skewed toward the parent substance, similar to the results from the spleen sample, although the ratio is not as strongly shifted toward methadone. 

This case marks the first instance in which this combination of drugs has been detected in *Ch. albiceps* larvae. Previous laboratory studies focused on the detection of a combination of three opioids (morphine, codeine, and methadone) and their metabolites in another calliphorid species [29], while others have explored the effects of methadone alone on the calliphorid development [19] and insect succession patterns [30]. These studies have revealed significant differences in blow fly development in the presence of methadone [19], and while the overall pattern of insect succession remains similar between carcasses treated with methadone versus untreated ones, the specific patterns of succession of *Ch. albiceps* and *Calliphora vicina* Robineau-Desvoidy (Diptera: Calliphoridae) differ slightly between the two [30].

These findings suggest that when the presence of methadone is confirmed in a cadaver colonized by insects, these can be used for toxicological analyses but should not be considered for PMI estimation. This underscores the need for further research on PMI correction factors in the presence of varying amounts of methadone in the insect food source.

Lastly, the larvae sample showed the presence of a small amount of benzoylecgonine (BEG), not detected in the human matrices analyzed. Benzoylecgonine, the main metabolite of cocaine and heroin, is typically considered a hydrophilic substance, and this characteristic could affect the distribution patterns that occur in humans and larvae [31]. Previous studies that explored the effects of cocaine and heroin’s main metabolites on the development rate of another calliphorid species [32] concluded that they significantly impacted insects’ development, suggesting that for these substances as well, insects found to be positive should not be used for PMI estimations.

Case 3: The toxicological analyses on the collected larvae yielded negative results, consistent with the analyses on the blood sample. The cause of death was determined to be mechanical asphyxia resulting from hanging, conducted with suicidal intent. Considering the manner of death and circumstantial evidence, comprehensive data collection (tissue samples and insect samples) ensured no aspect of the investigation was overlooked. Insects could have also been used to support the estimation of the postmortem interval (PMI); however, due to the limitations of the requests from the public prosecutor, environmental data was not provided.

Case 4: Due to the skeletonization of the body, in this case, the toxicological analyses were performed exclusively on the carrion insects, which showed positive results for three different drugs. This information confirmed the circumstantial data that the subject was prescribed a therapy with flurazepam, quetiapine, and lorazepam more than a year before her death. The larvae, in fact, showed the presence of quetiapine and desalkyl-flurazepam, which is a metabolite of flurazepam. The finding of a high concentration of pregabalin in the larvae, not listed in the therapy, and the lack of lorazepam suggests that the subject underwent a change of therapy. While various benzodiazepines [33,34], including lorazepam [35], have been considered in previous laboratory-based entomotoxicological studies, and case reports have identified both lorazepam and pregabalin in entomological samples [31], this case represents the first time this combination of drugs has been reported in larvae. As a consequence, in this case insects could not be considered for PMI estimation, and research is needed to further understand the effect of drug combinations on the survival and growth rate of necrophagous insects.

Entomological samples are particularly useful for toxicological analyses, both for technical reasons and the application of the results. Beginning with the scene investigation, entomological sampling is fast and easy, as the operator only needs a Falcon tube or a specimen cup and a spoon, tweezers, or just hands protected by gloves. Most common specimens to be collected include larvae, full or empty puparia [22]. Regarding the larvae, it is preferable to collect larvae of large size, when they have reached at least 0.5 cm in length or more, indicating that they have reached the third instar and have spent enough time feeding on the body to accumulate toxicological substances. Larvae move slowly and are often in large masses, making it easy to collect a large number in a single scoop. Full and empty puparia can be found on, under, and surrounding the body, and they do not move. The amount of specimens needed is limited, as only 1 g is required for the analysis. In contrast with specimens collected for morphological analyses, which require them to be sacrificed in hot water and stored in 70% ethanol [22], once collected, insect specimens intended for toxicological analyses must be stored at a temperature of −20 °C, after being carefully washed in water to remove debris from the environment and human remains. However, the cleaning step can also be performed before the analyses, which can be conducted after a short or long time from collection.

Regarding laboratory procedures, larvae have been reported to present a lower matrix effect for specific substances, such as methadone and EDDP, in certain studies [19]. However, the matrix effect can vary depending on the composition of the matrix, sample preparation, and chromatographic analysis. Overall, larvae seem to be an appropriate matrix for toxicological analyses, especially when specimens from the cadaver are not available. Moreover, the entomological sample is highly sensitive and can provide optimal qualitative analysis of the toxicological substances present in the feeding substrate. In the four cases presented here, comparing results obtained from larvae with those obtained from available human matrices, almost all substances present in the human matrices were consistently found in the entomological sample, and substances not found in human matrices were detected (e.g., oxazepam in case 1 or BEG in case 2). This highlights the fact that insects can be used to detect the presence of substances taken acutely or, in any case, not metabolically eliminated—unlike the hair matrix, which is known instead to indicate past consumption. 

To date, little is known about the quantitative value of the entomological sample, likely due to the already described wide variability of the sample and the environment in which insects populate. Small sample sizes and low analytical replication are additional factors that contribute to difficulties in interpretation. 

The results presented in the current four cases do not allow for inferring the dose of substance exposure. The only evidence that emerged from analyzing cases 1 and 2 was the maintenance of a substance/metabolite ratio skewed towards the unchanged substance, compared to the human matrix. This was observed in relation to codeine and methadone, respectively. This evidence does not hold true for diazepam and its metabolites detected in case 1: while in muscle tissue the diazepam/nordiazepam ratio was skewed towards nordiazepam, the opposite was observed in the larvae.

## 5. Conclusions

The results obtained in the four cases presented complement the scant literature currently available on the application of entomotoxicology in real cases [28,33,34,35,36], as compared to the large number of laboratory-based entomotoxicological studies in which researchers have used different animal models (rabbits and rats) or meat spiked with substances of various types, mixtures, and concentrations [8,15,29,33,37,38,39,40,41,42,43,44,45,46].

It is important to underline that the results presented here are derived from only four cases; therefore, it is not possible to make any statistical inference. However, in these cases, entomological samples have proven to be a valid substitute or complementary matrix to the body remains. Overall, entomotoxicological analyses should always be considered in forensic cases as a valuable method/complementary method for the qualitative identification of exogenous substances present in the body of the victim, as they can be crucial for guiding hypotheses about the causes of death, such as in cases of poisoning. 

Furthermore, entomotoxicological analyses should always be performed when circumstantial data suggest the use of drugs by the victim, in order to ensure that the PMI estimation is carried out correctly. Lastly, further studies should be conducted with the aim of producing correcting factors for the insect table of growth in the presence of toxicological substances. Considering the availability of such data in the future, it becomes pivotal during the investigation process to always acquire the micro/macro-environmental temperature at the scene and to perform an accurate identification of the insect species colonizing the remains. These data would significantly impact the accuracy of PMI estimations, especially when reopening a case in light of new entomotoxicological research discoveries.

## Figures and Tables

**Figure 1 insects-15-00500-f001:**
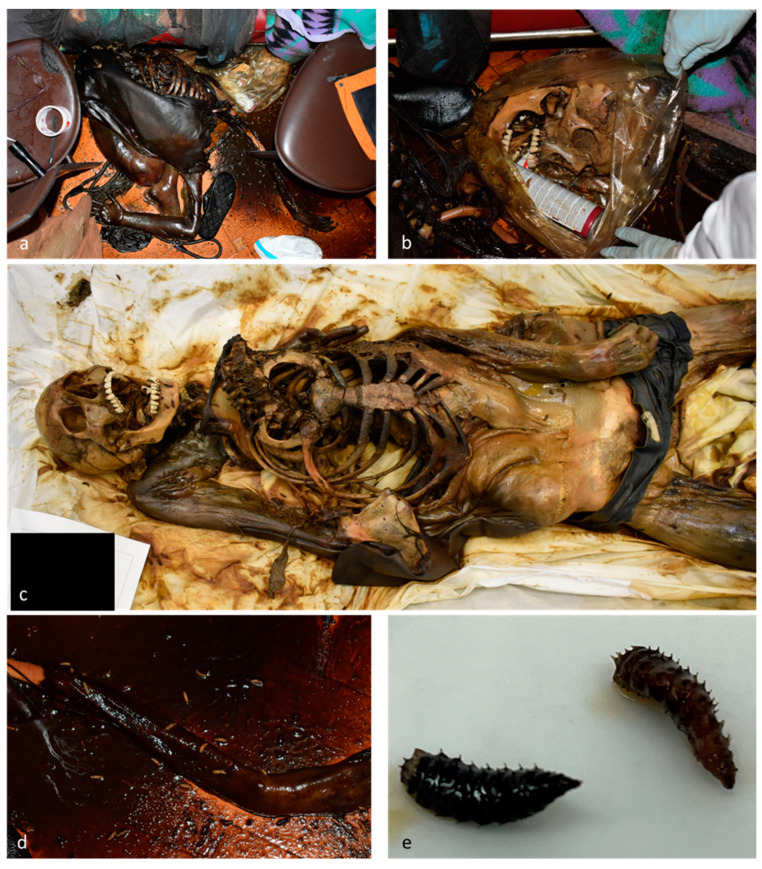
Case 1. (**a**,**b**) Scene of death; (**c**) external examination; (**d**) presence of larvae on the body and surroundings; (**e**) third instar larvae of *Chrysomya albiceps* (Diptera: Calliphoridae).

**Figure 2 insects-15-00500-f002:**
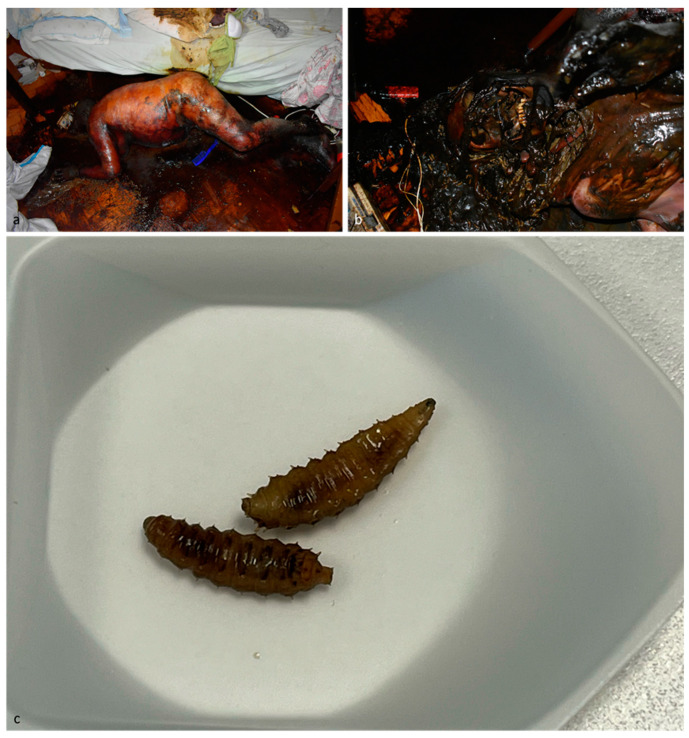
Case 2. (**a**,**b**) Scene of death; (**c**) third instar larvae of *Chrysomya albiceps* (Diptera: Calliphoridae).

**Figure 3 insects-15-00500-f003:**
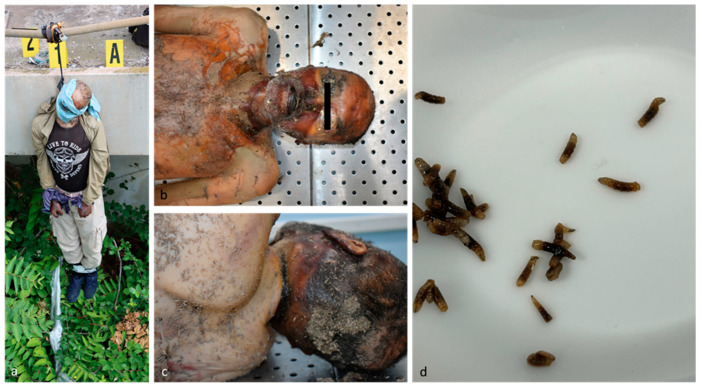
Case 3. (**a**) Scene of death; (**b**,**c**) external examination; (**d**) third instar larvae of calliphorids.

**Figure 4 insects-15-00500-f004:**
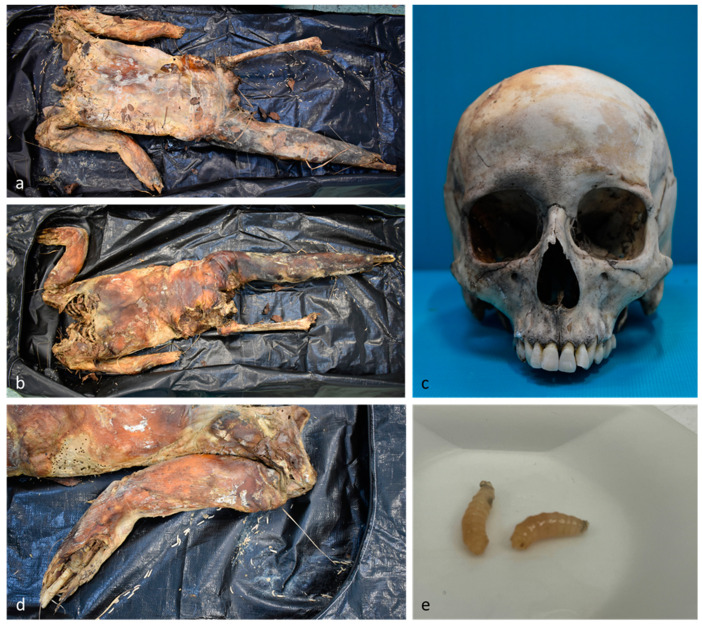
Case 4. (**a**–**d**) External examination; (**e**) third instar larvae of calliphorids.

**Table 1 insects-15-00500-t001:** Demographic characteristics, postmortem findings based on decomposition changes and last known sighting, and details regarding the samples used for toxicological analyses.

Case #	Gender, Age	Location	Circumstantial Data		Decomposition Changes	Estimated PMI Based on Decomposition Changes and Last Known Sighting	Toxicological Analyses Performed on	
			Death Scenario	Presence of Drugs			Human Specimens	Entomological Samples
1	Male, 59	Home	Presence of a plastic bag around the cadaver’s head, also containing a butane can. In the house, open kitchen gas knobs.	Numerous blister packs of zolpidem, gabapentin, duloxetine, levosulpiride, propantheline bromide/bromazepam, paracetamol/codeine phosphate.	Advance putrefactive phenomena in the chromatic-colliquative phase. Skeletonization of the upper part of the body.	20–30 days	Hair, muscle	*Ch. albiceps* L3, PF
2	Female, 58	Home	Subject in substitutive treatment with methadone.	Next to the cadaver presence of a blister of quetiapine and three bottles of methadone.	Advanced putrefactive phenomena in the chromatic-colliquative phase and skeletonization. Partial mummification.	Approximately 3 weeks	Spleen	*Ch. albiceps* L3
3	Male, 54	Suburban area	Homeless hanged by a rope at the parapet of a road. No signs of violence.	None associated with the body or the surrounding environment.	Advanced putrefactive phenomena in the chromatic-colliquative phase. Corification.	6–8 days	Blood	Calliphorids L3
4	Female, 67	Wood	Woman disappeared approximately 3 months before. Reported to be in treatment with flurazepam, quetiapine and lorazepam.	None associated with the body or the surrounding environment.	Advanced putrefactive phenomena- skeletonization. Special transformative phenomena: mummification and saponification.	2–3 months	n.a.	Calliphorids L3

L3 = third instar larvae; PF = post-feeding larvae; n.a. = not available.

**Table 2 insects-15-00500-t002:** Results of qualitative and quantitative toxicological investigations in samples of case 1.

Matrix	Results	Drugs									
		Coc.	Cod.	Norcod.	Morph.	Brom.	Diaz.	Nordiaz.	Temaz.	Oxaz.	Zolp.
Skeletal muscle	Screening	*n.d.*	+	+	+	+	+	+	+	*n.d.*	+
	Concentration (ng/mg)	/	3820	1311	331	434	6824	9654	781	/	6276
Hair	Screening	+	+	*n.d.*	+	*n.d.*	*n.d.*	*n.d.*	*n.d.*	*n.d.*	+
	Concentration (ng/mg)	0.03	0.91	/	0.09	/	/	/	/	/	0.41
Larvae	Screening	*n.d.*	+	+	+	+	+	+	+	+	+
	Concentration (ng/g)	/	235	113	28	3	58	24	7	12	18

Coc.: cocaine; Cod.: codeine; Norcod.: norcodeine; Morph.: morphine; Brom.: bromazepam; Diaz.: diazepam; Nordiaz.: nordiazepam; Temaz.: temazepam; Oxaz.: oxazepam; Zolp.: zolpidem. n.d. = not detected. + = positive. / = not investigated because it was not detected during screening.

**Table 3 insects-15-00500-t003:** Results of qualitative and quantitative toxicological investigations in samples of case 2.

Matrix	Results	Drugs				
		7-Aminoclonazepam	BEG	Methadone	EDDP	Quetiapine
Spleen	Screening	+	*n.d.*	+	+	+
	Concentrations (ng/g)	706	/	27,699	1990	3693
Larvae	Screening	+	+	+	+	+
	Concentrations (ng/g)	4	5	172	101	13

n.d. = not detected. + = positive. / = not investigated because it was not detected during screening.

**Table 4 insects-15-00500-t004:** Results of qualitative and quantitative toxicological investigations in samples of case 4. Due to the condition of the body, no other matrices besides larvae were collected or available for toxicological analyses.

Matrix	Results	Drugs		
		Desalkyl Flurazepam	Pregabalin	Quetiapine
Larvae	Screening	+	+	+
	Concentrations (ng/g)	86.8	456.0	14,623.0

+ = positive.

## Data Availability

All relevant data are available from the text, figures, and tables.

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
