# Peer review of "Comparative Toxicological Analyses of Traditional Matrices and Blow Fly Larvae in Four Cases of Highly Decomposed Human Cadavers"

_insects, 2024, doi:10.3390/insects15070500_

Round 1

Reviewer 1 Report

Comments and Suggestions for Authors

COMPARATIVE TOXICOLOGICAL ANALYSES OF TRADITIONAL MATRICES AND CARRION INSECTS IN FOUR CASES OF HIGHLY DECOMPOSED HUMAN CADAVERS.

General comment 1: this is a necesary paper in order to get a database about drugs and sarcosaprophagous díptera related to human corpses, but we must take in account that the cases are completely different (2 inside home, 1 into the Woods, 1 suburban área) no data about environmental conditions (temperaturas, rain, wind and humidity, etc) are given in the paper. I think is not posible to compare among them. All corpses are in advanced stage of decay but in 2 cases larvae is not accured identificated, Calliphoridae L3, and in the other Ch. albiceps L3. In table 1, in cases 3 y 4 there were no presence of drugs around corpses but in 4 the victim has a treatment with some drugs.

General comment 2:  In the paper is said that the analyzed simples were 1 grs of larvae, this could be a gold standard in medicine for human tissues, but 1 gr. could be a estándar for larvae?. Authors could give the number of analyzed larvae, and the number of larvae observed and sampled. Because, it could be less accourate when we are working with larvae díptera. Specially for, if 1 gr of larvae is a representative sample of what there was related to the corpse.

General comment 3: In case 3, in absence of drugs, it is clearly seen that authors could include not only PMI form the pathologist but include the PMI obtainde in the base of entomological evidences, with environmental conditions, to compare both results.

Comment 4: It is crucial research in the efecto of each drug isolated, in the larval development, but it is crucial too, to investigate how the drug mix could affect them. So by now, I agree with authors than, with positive toxicology analyses researchers must be cautious with the developmental rate in díptera larvae to calculate the PMI.

Line 96: when authors say “that Calliphoridae larvae are asociated with advanced stages of decay”. This sentence needs a description more accurate, because it depends on the climatic area, Third instar larvae can be found in earlier stages, due to climatic characteristics. I think authors have been done a general concept but it is not absolutely true.

Line 123: Materials and Methods. In my view, there are exposed 4 cases, but, authors should elimitate Case 3 due to negative results for all drugs. So this is not relevant for results; and paper Title is specific with toxicologícal analyses.

Line 150: The PMI is obtained by the pathologist, but it could be interesting to prospect if with the isects obtained a forensic entomologist could give a PMi of the same corpse. I think, authors should include that PMI calculated are based only in pathologist results. By the contrary, authors include that insects could not be used to get the PMI if drugs are found in the body. There are some papers on the effect of drugs in development of some species of larvae. This papers should be mentioned, althoug the target species is not Ch. albiceps. I am not agree with authors in this subject. Insects are evidences and could help with PMI. More research on the subject is needed, but in my view is usefulness.

Line 146-147: Authors can specifically explain where were caught the insects on the body, in all Surface, only in skeletoniced áreas, etc…

Line 153: Figure 1, figure d could not see clearly the larvae and where were them.

Line 157: please, specify the method for calculate PMI is based in indirect method, as eyewitness, in table 1 is said 2-3 weeks, and in the text “aproximately 3 weeks”.

Line 180: If authors want to keep the case 3, in this line, authors mention “between rainy and sunny days”. Temperature, rain and humidity and in addition sun exposition of a corpse can vary and severely the larvae development. This point must be taken in account.

Line 188:  In case 3, only third instar calliphorid larvae where observed. Was it impossible to identify?  And why? It is crucial to know why It was due to bad state of preservation? If is this the situation, results can be considered as accurate? Or no entomologist were consulted?.

Line 199: the same comment than in line 188

Line 293 to 309: Tables 1 to 4: It could be interesting to compare statistically the results obtained from human tissues and from larvae  for each drug founded. In order to know if 1 gr of larvae could be the gold starndard or must be analized other larvae weigth to be more accourate in next papers.

Author Response

Reviewer 1

General comment 1: this is a necessary paper in order to get a database about drugs and sarcosaprophagous díptera related to human corpses, but we must take in account that the cases are completely different (2 inside home, 1 into the Woods, 1 suburban área) no data about environmental conditions (temperaturas, rain, wind and humidity, etc) are given in the paper. I think is not possible to compare among them. All corpses are in advanced stage of decay but in 2 cases larvae is not accured identificated, Calliphoridae L3, and in the other Ch. albiceps L3. In table 1, in cases 3 y 4 there were no presence of drugs around corpses but in 4 the victim has a treatment with some drugs.

Author: The reviewer has a good point considering the best practice in forensic entomology, which requires collecting micro and macro-environmental information alongside insect samples and identifying the collected species to the species level (see Bambaradeniya et al, 2023). However, this practice is necessary only when insects are used for the estimation of the time since death. In the cases discussed, the public prosecutor in charge of the investigation did not request such an estimation; therefore, environmental data were not collected, and specimens did not undergo species identification.

In the originally submitted paper, it was clarified that the cases are “heterogeneous both for age and sex of the subjects, as well as time of the year and environmental conditions.” In accordance with the suggestions of the reviewers, “and presence of drugs in the surrounding environment” has been added to the sentence.

In the originally submitted paper, it was clarified that even with the correct species identification and the environmental data, the estimation of the PMI based on the insects would not have been possible due to the lack of research on the effect of the observed drugs/cocktail of drugs on carrion insects. Thanks to the reviewer’s comment, the text has now been improved to provide clarification regarding the public prosecutor's request, and the conclusions have been improved to emphasize the importance of collecting such data.

General comment 2:  In the paper is said that the analyzed simples were 1 grs of larvae, this could be a gold standard in medicine for human tissues, but 1 gr. could be a estándar for larvae?. Authors could give the number of analyzed larvae, and the number of larvae observed and sampled. Because, it could be less accourate when we are working with larvae díptera. Specially for, if 1 gr of larvae is a representative sample of what there was related to the corpse.

Author: 1g of larvae is the amount used for toxicological analyses, and it is an amount that has been validated by several entomotoxicological studies. The number of larvae is not important because different species at different ages weigh differently. For example, 1g of third instar larvae of L. sericata can be made up of 5-15 larvae (depending on the L3 stage and the specimen’s food intake), while the same species in L1 or L2 would require 100+ specimens. Since this is common knowledge in entomology, forensic entomology, and entomotoxicology practice, no modifications are applied to the original text.

General comment 3: In case 3, in absence of drugs, it is clearly seen that authors could include not only PMI form the pathologist but include the PMI obtainde in the base of entomological evidences, with environmental conditions, to compare both results.

Author: The reviewer has a good point, but it is necessary to consider that as discussed in the first comment, experts working on a case can only provide answers based on the request of the public prosecutor, who in this case didn’t request the use of the entomological data for PMI estimation. Sometimes, unfortunately, it is a matter of investigation costs, and when the case is not high profile, certain analyses are not requested. Without being polemic against the justice system, the reviewer’s good point has not been added to the text.

Comment 4: It is crucial research in the efecto of each drug isolated, in the larval development, but it is crucial too, to investigate how the drug mix could affect them. So by now, I agree with authors than, with positive toxicology analyses researchers must be cautious with the developmental rate in díptera larvae to calculate the PMI.

Authors: we thank the reviewer. This point is raised in the conclusion.

Line 96: when authors say “that Calliphoridae larvae are asociated with advanced stages of decay”. This sentence needs a description more accurate, because it depends on the climatic area, Third instar larvae can be found in earlier stages, due to climatic characteristics. I think authors have been done a general concept but it is not absolutely true.

Authors: Without delving into further details, as this is an introduction for an entomotoxicology paper rather than a taphonomy paper, the word 'generally' has now been added to make the sentence more general. 

Line 123: Materials and Methods. In my view, there are exposed 4 cases, but, authors should elimitate Case 3 due to negative results for all drugs. So this is not relevant for results; and paper Title is specific with toxicologícal analyses.

Authors: Despite the negative results of the toxicological analyses, we believe Case 3 is important to report because the manner of death and circumstantial evidence (a medical report indicating a diagnosis of anxious-depressive syndrome with previous substance abuse) suggested that substances might have been present. This case highlights the importance of using insects in toxicological analyses to ensure no stones are left unturned during an investigation, particularly when a body is found in a highly decomposed state.

To clarify the reasons for including this case despite the negative results, we have added an explanation in the discussion section. This inclusion also supports the development of further guidelines for best practices and crime scene investigations. 

Line 150: The PMI is obtained by the pathologist, but it could be interesting to prospect if with the isects obtained a forensic entomologist could give a PMi of the same corpse. I think, authors should include that PMI calculated are based only in pathologist results. By the contrary, authors include that insects could not be used to get the PMI if drugs are found in the body. There are some papers on the effect of drugs in development of some species of larvae. This papers should be mentioned, althoug the target species is not Ch. albiceps. I am not agree with authors in this subject. Insects are evidences and could help with PMI. More research on the subject is needed, but in my view is usefulness.

Authors: As discussed in previous comments, experts working on a case can only provide answers based on the request of the public prosecutor, who in this case didn’t request the use of the entomological data for PMI estimation or the collection of environmental data or species ID. Without being polemic against the justice system, the reviewer’s good point has not been added to the text.

Line 146-147: Authors can specifically explain where were caught the insects on the body, in all Surface, only in skeletoniced áreas, etc…

Authors: this has been clarified in the text

Line 153: Figure 1, figure d could not see clearly the larvae and where were them.

Authors: the presence of larvae is pretty clear in the picture 1d, it is concerning if the reviewer is not able to spot them. 

Line 157: please, specify the method for calculate PMI is based in indirect method, as eyewitness, in table 1 is said 2-3 weeks, and in the text “aproximately 3 weeks”.

Authors: this has been amended and clarified in the text

Line 180: If authors want to keep the case 3, in this line, authors mention “between rainy and sunny days”. Temperature, rain and humidity and in addition sun exposition of a corpse can vary and severely the larvae development. This point must be taken in account.

Authors: As discussed previously, experts involved in a case can only provide answers based on the requests of the public prosecutor. In this instance, the prosecutor did not request the use of entomological data for post-mortem interval (PMI) estimation or the collection of environmental data or species ID. However, we acknowledge the comment regarding the impact of rain on carrion insects. Notably, one of the authors of this paper has conducted research on the effects of submergence of carrion insects in various types of water, and a reference to this research has been included.

Line 188:  In case 3, only third instar calliphorid larvae where observed. Was it impossible to identify?  And why? It is crucial to know why It was due to bad state of preservation? If is this the situation, results can be considered as accurate? Or no entomologist were consulted?.

Authors:  As discussed previously, experts involved in a case can only provide answers based on the requests of the public prosecutor. In this instance, the prosecutor did not request the use of entomological data for post-mortem interval (PMI) estimation or the collection of environmental data or species ID. This has been now clarified in the text at the beginning of the discussion of the cases.

Line 199: the same comment than in line 188

Authors:  As discussed previously, experts involved in a case can only provide answers based on the requests of the public prosecutor. In this instance, the prosecutor did not request the use of entomological data for post-mortem interval (PMI) estimation or the collection of environmental data or species ID. This has been now clarified in the text at the beginning of the discussion of the cases.

Line 293 to 309: Tables 1 to 4: It could be interesting to compare statistically the results obtained from human tissues and from larvae for each drug founded. In order to know if 1 gr of larvae could be the gold starndard or must be analized other larvae weigth to be more accourate in next papers.

Authors: Thank you. This is an interesting future avenue for research, although this comment does not need to be included in the current text.

Reviewer 2 Report

Comments and Suggestions for Authors

Line

Comment

21

Insects cannot detect the specimens. They serve as the substrate from which toxicological analysis may occur. Wording needs refinement

62

“none” is not the correct word

78-87

A long-winded sentence that needs to be broken down for readability

79

I would add biomagnification here

103

Repetitive from above - unnecessary

109

“On the contrary, other studies have shown a correlation between 109 drugs in substrates and larvae samples. “ Such as? There should be references provided here to follow on from the preceding statement

145, 149, 166, 392, 399

Italicise species name

169

“sample was taken” – word missing

308

This should indicate no matrix was collectable 

319

This is debatable – there is a large body of work available, and this should be acknowledged more fully

399

Please assign authority at first introduction of a species name

446

This sentence is unclear “larvae seem to be a better matrix compared to 446 specimens taken from cadavers”

The study presents itself as novel and filling an empty niche, but the use of insects as targets for detection of drugs has been performed repeatedly. However, quantification has also been refuted repeatedly – there are some outstanding recent reviews, as well as research studies that have demonstrated this, and should be presented more openly in this manuscript.

The introduction is superficial in places and lacking critical consideration – there are very good reasons why quantification (and in many cases qualification) has received limited support from the forensic entomological community. The great propensity for false negative results, and results that reflect misleading levels of toxicants, is ignored in this introduction and needs to be addressed. While systematic studies that utilise toxicants homogenised through mince meat appear a poor substitute, they are in fact the most controlled, ethical option available in the field. These allow the scientist to ensure even exposure to substance, ensuring they have ingested it. Ideally a whole organism model is preferable as it allows for administration of the substance premortem and thus downstream studies may also reflect metabolites, but human cases suffer from uneven distribution of substance in the body following death, based on dose, nature of toxicant, method of administration, time elapsed between administration and death, individual health factors, chronic vs acute exposure to the substance, interactions between multiple simultaneously present substances, as finally, the effect of feeding site of the insect. Studies investigating cadavers are extremely limited in their ability to control so many factors and are a risky basis from which to support quantification. The PMR is acknowledged in the discussion – so why not earlier?

Chrysomya albiceps is a predatory species. How do you know these larvae had not fed on other larvae of earlier colonising specie, resulting in potential biomagnification? This should be explored in discussion – the behaviour of this species complicates its potential use. Nuroteva & Nuorteva demonstrated the effects of biomagnification with beetles feeding on larvae of Protophormia terranovae that ingested mercury.

Results should be stated, not just presented in tables – key findings should be drawn out and articulated.

While the study draws some nice observations, it isn’t clear what overall stance is reached regarding quantification. It is wise to be cautious and circumspect with such low replication, but the early framing of the paper to indicate this would be discussed seems to be left adrift – the paper needs tightening in this respect. Further, the statements regarding methadone and mixtures of drugs are somewhat overstated based on small sample size.

The collection methodology itself is not novel – it would be interesting to advise whether SOPs for collection of insect specimens need wide revision in light of common practise of HWK then ethanol storage – does this preclude entomotoxicological analysis?

 The observations are of interest, but I believe the authors can make more valid commentary here that considers the entomological limitations as affected by the species and their behaviour with regard to human remains more accurately. The determination of conclusions requires more holistic consideration.

Author Response

Reviewer 2

Line

Comment

21

Insects cannot detect the specimens. They serve as the substrate from which toxicological analysis may occur. Wording needs refinement

Authors: this has been amended

62

“none” is not the correct word

Authors: this has been amended

78-87

A long-winded sentence that needs to be broken down for readability

Authors: this has been amended

79

I would add biomagnification here

Authors: this has been added, thank you for rasing this point.

103

Repetitive from above – unnecessary

Authors: this has been amended

109

“On the contrary, other studies have shown a correlation between 109 drugs in substrates and larvae samples. “ Such as? There should be references provided here to follow on from the preceding statement

Authors: this has been amended

145, 149, 166, 392, 399

Italicise species name

Authors: this has been amended

169

“sample was taken” – word missing

Authors: the sentence has been amended

308

This should indicate no matrix was collectable 

Authors: this has been amended and clarified in the text.

319

This is debatable – there is a large body of work available, and this should be acknowledged more fully

Authors: the sentence has been amended to highlight the limited literature on cases as compared to research. The paper of importance are reported throughout the discussion.

399

Please assign authority at first introduction of a species name

Authors: this has been amended

446

This sentence is unclear “larvae seem to be a better matrix compared to 446 specimens taken from cadavers”

 Authors: this has been amended

R2: The study presents itself as novel and filling an empty niche, but the use of insects as targets for detection of drugs has been performed repeatedly. However, quantification has also been refuted repeatedly – there are some outstanding recent reviews, as well as research studies that have demonstrated this, and should be presented more openly in this manuscript.

Authors: all the recent papers have been acknowledged throughout the manuscript, and the limitations of old and new works have been stated.

R2: The introduction is superficial in places and lacking critical consideration – there are very good reasons why quantification (and in many cases qualification) has received limited support from the forensic entomological community. The great propensity for false negative results, and results that reflect misleading levels of toxicants, is ignored in this introduction and needs to be addressed. While systematic studies that utilise toxicants homogenised through mince meat appear a poor substitute, they are in fact the most controlled, ethical option available in the field. These allow the scientist to ensure even exposure to substance, ensuring they have ingested it. Ideally a whole organism model is preferable as it allows for administration of the substance premortem and thus downstream studies may also reflect metabolites, but human cases suffer from uneven distribution of substance in the body following death, based on dose, nature of toxicant, method of administration, time elapsed between administration and death, individual health factors, chronic vs acute exposure to the substance, interactions between multiple simultaneously present substances, as finally, the effect of feeding site of the insect. Studies investigating cadavers are extremely limited in their ability to control so many factors and are a risky basis from which to support quantification. The PMR is acknowledged in the discussion – so why not earlier?

Authors: The focus of the paper is to discuss cases, rather than delve deeply into a polemic regarding the reasons why certain studies haven’t been performed (generally due to lack of funds) or the issues caused by different experimental designs (sometimes due to animal ethics permissions not being granted in institutions of certain countries). While all these points have been addressed in discussion for the overall balance of the manuscript, the aim is to maintain focus and report relevant literature. Readers interested in specifics of experimental design, issues of quality/quantity, and research interest can find detailed discussions in the relevant literature.

R2: Chrysomya albiceps is a predatory species. How do you know these larvae had not fed on other larvae of earlier colonising specie, resulting in potential biomagnification? This should be explored in discussion – the behaviour of this species complicates its potential use. Nuroteva & Nuorteva demonstrated the effects of biomagnification with beetles feeding on larvae of Protophormia terranovae that ingested mercury.

Authors: Due to the environmental conditions, Chrysomya albiceps was the only species present at the scene. Since no research has been conducted on the biomagnification of Chrysomya albiceps feeding on other calliphorid species, such as Lucilia and Calliphora, which sometimes arrive at the body before it, any comments would be speculative and are not necessary for the purpose of this paper.

R2: Results should be stated, not just presented in tables – key findings should be drawn out and articulated.

Authors: Based on previous publishing experience with this and other journals, presenting results twice (in both table and text formats) would be redundant. Key findings are reported in the Results section and discussed in the Discussion section.

R2: While the study draws some nice observations, it isn’t clear what overall stance is reached regarding quantification. It is wise to be cautious and circumspect with such low replication, but the early framing of the paper to indicate this would be discussed seems to be left adrift – the paper needs tightening in this respect. Further, the statements regarding methadone and mixtures of drugs are somewhat overstated based on small sample size.

Authors: We agree with the comment of R2, and we have further clarified and contextualized our results at the end of the discussion. Some points are already present in the conclusion and are not repeated to avoid redundancy (“Overall, entomotoxicological analyses should always be considered in forensic cases as a valuable method/complementary method for the qualitative identification of exogenous substances present in the body of the victim”.)

R2: The collection methodology itself is not novel – it would be interesting to advise whether SOPs for collection of insect specimens need wide revision in light of common practise of HWK then ethanol storage – does this preclude entomotoxicological analysis?

Authors: The collection methodology in entomotoxicology contexts is not fully detailed in the most popular paper on best practices in forensic entomology (Amendt et al., 2007), and is only briefly touched upon in the most recent paper (Bambaradeniya et al., 2023). Both of these papers were authored by forensic entomologists, whereas ours involves a team of forensic pathologists, toxicologists, and forensic entomologists. The intended audience for our paper is diverse, and we believe that practical guidelines drawn from real case experiences will be more effective than those presented in previous works (as also mentioned).

The preservation methods (hot water and ethanol) mentioned in the existing literature do not preclude subsequent toxicological analyses, but they may affect the results. Further research should investigate the extent of these alterations, whether they are influenced by insect age, time elapsed, or the type/amount of drug present. The paper now mentioned in this section provides explanations on this topic

R2: The observations are of interest, but I believe the authors can make more valid commentary here that considers the entomological limitations as affected by the species and their behaviour with regard to human remains more accurately. The determination of conclusions requires more holistic consideration.

Authors: A more holistic consideration would have required different baseline data, such as detailed species identification and assessment of the environmental situation. However, in our system, experts involved in a case can only provide answers based on the requests of the public prosecutor. In this instance, the prosecutor did not request the use of entomological data for post-mortem interval (PMI) estimation, collection of environmental data, or species identification. This clarification has now been included at the beginning of the discussion of the cases. Discussing unpresented data would be speculative or polemic, detracting from the aim of the paper.

Reviewer 3 Report

Comments and Suggestions for Authors

A paper well worth publication which I fully endorse. It fills a long vacant gap in comparative research and surely represents a foundation for future research. 

There are a few issues that need to be addressed before the paper can be published however. There are marked in green in the attached text. 

I inadvertently placed a bookmark in page 2, which I cannot remove - my apologies.

Comments on the Quality of English Language

The standard of English is good, but there is an overuse of the word 'however’. Despite common usage, beginning a sentence with a conjunctive is grammatically poor. A conjunctive should be used to join two phrases, so in this instance it could be placed after ‘controversial’ and between commas, that is ‘…controversial, however, due to …’’. In some instances the use has been inappropriate applied and I have indicated where these instances are.

Author Response

The standard of English is good, but there is an overuse of the word 'however’. Despite common usage, beginning a sentence with a conjunctive is grammatically poor. A conjunctive should be used to join two phrases, so in this instance it could be placed after ‘controversial’ and between commas, that is ‘…controversial, however, due to …’’. In some instances the use has been inappropriate applied and I have indicated where these instances are.

Authors: the paper has been amended following the suggestions of R3
